# Comparing the Efficacy of Intra-Articular Single Platelet-Rich Plasma(PRP) versus Novel Crosslinked Hyaluronic Acid for Early-Stage Knee Osteoarthritis: A Prospective, Double-Blind, Randomized Controlled Trial

**DOI:** 10.3390/medicina58081028

**Published:** 2022-08-01

**Authors:** Ying-Chun Wang, Chia-Ling Lee, Yu-Jen Chen, Yin-Chun Tien, Sung-Yen Lin, Chung-Hwan Chen, Paul Pei-Hsi Chou, Hsuan-Ti Huang

**Affiliations:** 1Ph.D. Program in Biomedical Engineering, College of Medicine, Kaohsiung Medical University, Kaohsiung 807, Taiwan; ycwang.ow@gmail.com; 2Department of Orthopedics, Kaohsiung Medical University Hospital, Kaohsiung Medical University, Kaohsiung 807, Taiwan; d740113@kmu.edu.tw (Y.-C.T.); sungyenlin@kmu.edu.tw (S.-Y.L.); hwan@kmu.edu.tw (C.-H.C.); 3Department of Orthopedics, Kaohsiung Municipal Ta-Tung Hospital, Kaohsiung Medical University, Kaohsiung 807, Taiwan; 4Department of Orthopedics, Kaohsiung Municipal Hsiao-Kang Hospital, Kaohsiung Medical University, Kaohsiung 807, Taiwan; 5Department of Sports Medicine, College of Medicine, Kaohsiung Medical University, Kaohsiung 807, Taiwan; 6Department of Physical Medicine and Rehabilitation, Kaohsiung Medical University Hospital, Kaohsiung Medical University, Kaohsiung 807, Taiwan; cathyleetw@gmail.com; 7Graduate Institute of Medicine, College of Medicine, Kaohsiung Medical University, Kaohsiung 807, Taiwan; 8Department of Physical Medicine and Rehabilitation, School of Medicine, College of Medicine, Kaohsiung Medical University, Kaohsiung 807, Taiwan; 9Department of Radiology, Kaohsiung Medical University Hospital, Kaohsiung Medical University, Kaohsiung 807, Taiwan; 980786@kmuh.org.tw; 10Department of Orthopedics, School of Medicine, College of Medicine, Kaohsiung Medical University, Kaohsiung 807, Taiwan; 11Regeneration Medicine and Cell Therapy Research Center, Kaohsiung Medical University, Kaohsiung 807, Taiwan; 12Orthopedic Research Center, Kaohsiung Medical University, Kaohsiung 807, Taiwan

**Keywords:** osteoarthritis, hyaluronic acid, platelet-rich plasma, cartilage, growth factor

## Abstract

*Background and Objectives*: For the treatment of knee osteoarthritis (OA), intra-articular platelet-rich plasma (PRP) and novel crosslinked single-dose hyaluronic acid (HA) have both been reported to improve outcomes, but no study has compared them for the treatment of knee OA. We hypothesized patients with early-stage knee OA who received PRP injections would have more WOMAC score changes than those who received HA injections. This is the first prospective, double-blind, parallel, randomized controlled trial comparing the efficacy of intra-articular single-dose PRP versus novel crosslinked HA (HyajointPlus) for treating early-stage knee OA. *Materials and Methods*: This study analyzed 110 patients randomized into the PRP (*n* = 54) or HA (*n* = 56) groups. The primary outcome is the change of WOMAC score at 1-, 3-, and 6-month follow-ups compared to baseline. *Results*: The data revealed significant improvements in all WOMAC scores in the PRP group at 1-, 3-, and 6-month follow-up visits compared with the baseline level except for the WOMAC stiffness score at the 1-month follow up. In the HA group, significant improvements were observed only in the WOMAC pain score for all the follow-up visits and in WOMAC stiffness, function, and total scores at 6-month follow-up. When comparing the change of WOMAC score at 1-, 3-, and 6-month follow-ups, no significant differences were found between PRP and HA group. *Conclusions*: This study revealed that both PRP and HA can yield significant improvements in WOMAC scores at 6-month follow-up without any between-group differences at 1-, 3-, and 6-month follow-ups. Thus, both the single-injection regimens of PRP and HA can improve the functional outcomes for treating early-stage knee OA.

## 1. Introduction

Knee osteoarthritis (OA) is a common degenerative joint disease characterized by loss of articular cartilage, subchondral bone changes, and inflammation of synovial tissue [1,2,3]. The major symptoms of knee OA are pain, loss of function, and swelling [4]. It mostly affects older adults (age ≥ 65 years), and it affects women (incidence: 18%) more than men (incidence: 9.6%), according to a 2004 study involving patients ≥60 years of age [5]. An estimated 12.4 million (33.6%) older adults in the US have knee OA [4]. Risk factors related to knee OA include obesity, older age, previous knee trauma, hand OA, and female sex [6]. Surgical interventions including arthroscopic surgery, osteotomy, and arthroplasty are considered the next steps for treating knee OA, after other nonsurgical treatments have been attempted, such as physical therapy [7], oral nonsteroidal anti-inflammatory drugs (NSAIDs) [8], and intra-articular injections (of platelet-rich plasma [PRP], hyaluronic acid [HA], and corticosteroids) [8,9,10].

Platelet-rich plasma (PRP) is autologous conditioned plasma containing a high concentration of platelets [11]. Platelets are a natural source of growth factors, such as insulin-like growth factor, platelet-derived growth factor, vascular endothelial growth factor, transforming growth factor beta, and platelet-derived angiogenic factor. Growth factors play a vital role in regenerative processes and wound healing [12]. The use of PRP in the treatment of knee OA has generated considerable attention [13].

Hyaluronic acid (HA) widely exists in human tissues, such as vitreous humor, heart valves, the umbilical cord, synovial fluid, skin, and skeletal tissues. Within the synovial fluid, HA provides joint lubrication, thus preventing the mechanical degradation of cartilage. In individuals with knee OA, HA synthesis and degradation are abnormal, resulting in reduced concentration and molecular weight of HA at the joint. These pathological changes reduce synovial fluid viscoelasticity, leading to cartilage damage; thus, intra-articular HA injection is a common therapy to treat knee OA. Some studies have mentioned better efficacy with a high molecular weight HA; however, some studies show no differences between high and low molecular weight HA. The optimal molecular weights, concentrations, and volumes of HA have not been established [14].

Several clinical trials and reviews have concluded that PRP injections have good clinical efficacy in the treatment of knee OA [15,16,17,18,19,20,21]. In addition, several trials and meta-analyses have reported that PRP injections are more effective than HA injections in terms of pain and physical function [22,23,24,25,26]. However, heterogeneity among studies was detected in their subgroup analyses, indicating large differences in clinical outcomes. Although the trials included in the meta-analysis were randomized controlled trials (RCTs), most of them (10 of 15) had a relatively small sample size (*n* < 50 for each group). Therefore, more high-level quality trials with a larger sample size are required to investigate the efficacy of PRP versus HA injections in the treatment of knee OA.

Conventional HA treatment for knee OA requires 3–5 intra-articular injections. However, newer HA products have longer-lasting activity, necessitating only one injection. An example of a new HA product is chemically crosslinked HA (e.g., HYAJOINT Plus), which results in increased viscoelasticity. It can significantly improve pain and functional results in knee OA [27,28,29].

Considering the aforementioned context, this study compared the efficacy of intra-articular single PRP versus novel crosslinked HA for the treatment of early-stage knee OA. We hypothesized patients with early-stage knee OA who received PRP injections would have more WOMAC score changes than those who received HA injections. This study provided robust evidence with a large sample size to assess the efficacy of PRP versus HA in treating knee OA. To our knowledge, ours is the first prospective, double-blind, randomized controlled trial comparing the efficacy of intra-articular single PRP versus novel crosslinked HA (HyajointPlus) for the treatment of early-stage knee OA.

## 2. Materials and Methods

### 2.1. Patient Selection

This prospective, double-blind RCT recruited patients visiting the outpatient orthopedic clinic of Kaohsiung Medical University Hospital from November 2017 to April 2019 if they met the following criteria: (1) age > 50 years; (2) a diagnosis of primary knee osteoarthritis; and (3) Kellgren–Lawrence (K–L) grade < 3. Patients with any one of the following criteria were excluded: (1) age < 50 years; (2) K–L grade ≥ 3; (3) history or active presence of clinically significant inflammatory articular or rheumatic disease other than OA; (4) rapidly progressive OA before the start of the trial; (5) history of lower extremity surgery; (6) excessive mechanical axis deviation (varus > 5°, valgus > 5°); (7) body mass index > 30; (8) history or presence of malignant disorders; (9) systemic disorders, such as diabetes mellitus, severe cardiovascular diseases, hematologic diseases, immune-deficiencies, and infections; (10) systematic or intra-articular corticosteroid therapy in the previous 3 months; (11) prior treatment with HA in the past 6 months; (12) anticoagulants or antiaggregant therapy in the preceding 30 days; (13) nonsteroidal anti-inflammatory medications in the preceding 7 days; (14) platelet count < 150,000/mL; or (15) hemoglobin < 12 g/dL. Glucosamine, chondroitin, analgesics, or physical therapy were not permitted during the study period. Acetaminophen (500 mg; maximum daily dose, 4 g) can be used as a rescue medication [27].

All patients provided informed consent before participation in the study. The study was conducted in accordance with the Declaration of Helsinki, and the protocol was approved by the Institutional Review Board of Kaohsiung Medical University Hospital, Kaohsiung, Taiwan (IRB NO: KMUHIRB-F(II)-20170062). The study was registered at ClinicalTrials.gov (NCT04972383).

### 2.2. Randomization and Masking

This is a prospective, double-blind, parallel randomized controlled trial. The included patients were randomly allocated to the PRP or HA group at a 1:1 ratio by using permuted block randomization with block sizes of two, four, and six. Randomization was concealed through the use of opaque and sealed envelopes until the treatment was assigned. Both patients and physicians were blinded to the treatment allocation.

### 2.3. PRP and HA Preparation

A single unit of 10 mL of peripheral venous blood was harvested from all participants before injection.

In the PRP group, an Aeon Acti-PRP (Aeon Biotherapeutics Corporation, 7F., No.2, Ln.258, Ruiguang Rd., Neihu Dist., Taipei City, Taiwan) set was used, whole blood was centrifuged at 3200 rpm for 6 min, and an approximate volume of 4 mL PRP was obtained for use in a single-dose treatment using a leukocyte-poor technique.

In the HA group, HYAJOINT Plus (3 mL, SciVision Biotech, No.9, S.6th Rd., Qianzhen Dist., Kaohsiung City, Taiwan) containing 60 mg of purified sodium hyaluronate was used for single-dose treatment. HYAJOINT Plus is a novel HA product comprising chemically crosslinked HA, resulting in increased viscoelasticity and durable activity and requiring only one injection. HYAJOINT Plus is produced by microbial fermentation and synthesized by a novel cross-linking process by 1,4-butanediol diglycidyl ether (BDDE) to introduce an antidegradation feature. The carefully controlled cross-linking creates a viscous gel with an increased density of HA (2% of HA, 20 mg/mL) [27].

Both PRP and HA are injected from the anterolateral portal of the knee, which is 1 cm above the joint line and just lateral to the patellar tendon, with 5cc syringe and 21G needle. All PRP and HA injection syringes were covered with opaque envelopes, and the intra-articular injection procedure was performed by the same physician without the use of guidance from ultrasound or other imaging techniques.

### 2.4. Outcome Assessment

To screen the patients, blood analysis was applied to each patient 1 week before the treatment. The main characteristics were also recorded at baseline, including age, sex, BMI, and K–L grading. The Western Ontario and McMaster Universities Index (WOMAC) score was used to assess the patients’ clinical outcomes at baseline and 1-, 3-, and 6-month follow-ups [30]. The WOMAC is a self-reported measure and contains three subscales, including the pain (5 items), stiffness (2 items), and physical function (17 items) subscales. The primary outcome is the change of WOMAC score at 1-, 3-, and 6-month follow-ups compared to baseline.

### 2.5. Statistical Analysis

The minimum requisite sample size was calculated using G*Power software (v.3.0.10, a free to use software from Heinrich-Heine-University Düsseldorf, Germany). The statistical tests included a between-group F-test and repeated-measures tests. The minimum sample size was calculated to be 31 participants per group (α = 0.05; power = 0.8; number of groups = 2; number of measurements = 4; because no preliminary data were available, we used moderate values of 0.25 for Cohen’s effect size *f* and 0.3 for Cohen’s effect size r of correlation among repeated measures). Assuming a 20% dropout rate, we estimated the minimum number of participants to be 38 per group. Categorical variables were expressed in terms of frequency or percentage, and their difference between the two groups at baseline were compared using an χ^2^ test. Continuous variables of age and BMI were expressed in terms of the mean ± standard deviation, and the differences between the two groups were compared using a two-sample t test. Continuous variables of WOMAC scores were expressed in terms of the mean ± standard error of the mean, and the differences between the two groups at baseline and follow-up visits were compared using a two-sample *t* test. The changes between baseline and follow-up visits and the changes between the two groups were assessed using generalized estimating equations. Statistical analyses were performed using IBM SPSS software (v.19; IBM, Armonk, NY, USA). Statistical significance was indicated at *p* < 0.05.

## 3. Results

The 116 included patients were randomly assigned to PRP and HA groups (*n* = 58 each). Among these patients, one patient did not receive the allocated treatment (the PRP group). Four patients dropped out of the trial (three from the PRP group and one from the HA group), and one was lost to follow-up (the HA group). Eventually, the data of 110 patients (54 in the PRP group and 56 in the HA group) were analyzed (Figure 1).

The mean ages in the PRP and HA groups were 61.87 ± 5.46 and 63.00 ± 5.33 years, respectively. Both groups had more women than men (77.8% and 71.4% women in the PRP and HA groups, respectively). Both groups did not differ with respect to age, sex, BMI, K–L grade, treatment side, and WOMAC scores at baseline (Table 1).

### 3.1. PRP Group

The mean WOMAC total score was 18.74 ± 1.85 at baseline and 14.96 ± 1.60, 13.43 ± 1.42, and 14.39 ± 1.56 at 1-, 3-, and 6-month follow-ups, respectively (Table 2). Relative to the baseline, the WOMAC total score significantly improved at the 1-month (−3.78 ± 1.20, *p* = 0.001), 3-month (−5.31 ± 1.48, *p* < 0.001), and 6-month follow-ups (−4.35 ± 1.61, *p* = 0.006). The WOMAC total score improved continually from baseline at 1-month and 3-month follow-ups but not at 6-month follow-up (Table 3 and Figure 2).

The mean WOMAC pain score was 4.20 ± 0.42 at baseline and 3.28 ± 0.37, 2.56 ± 0.32, and 2.87 ± 0.35 at 1-, 3-, and 6-month follow-ups, respectively (Table 2). Compared with the baseline, the WOMAC pain score significantly improved at the 1-month (−0.93 ± 0.33, *p* = 0.005), 3-month (−1.65 ± 0.38, *p* < 0.001), and 6-month (−1.33 ± 0.41, *p* = 0.001) follow-ups. The WOMAC pain score improved continually from baseline at 1-month and 3-month follow-ups but not at 6-month follow-up (Table 3 and Figure 2).

The mean WOMAC stiffness score was 1.89 ± 0.19 at baseline and 1.67 ± 0.19, 1.52 ± 0.16, and 1.37 ± 0.17 at the 1-, 3-, and 6-month follow-ups, respectively (Table 2). Compared with the baseline, the WOMAC stiffness score did not significantly improve at the 1-month follow-up (−0.22 ± 0.14, *p* = 0.107) but significantly improved at the 3-month follow-up (−0.37 ± 0.17, *p* = 0.026) and 6-month follow-ups (−0.52 ± 0.18, *p* = 0.004) (Table 3 and Figure 2).

The mean WOMAC function score was 12.65 ± 1.38 at baseline and 10.02 ± 1.15, 9.35 ± 1.03, and 10.15 ± 1.12 at the 1-, 3-, and 6-month follow-ups, respectively (Table 2). Compared with the baseline, the WOMAC function score significantly improved at the 1-month (−2.63 ± 0.96, *p* = 0.006), 3-month (−3.30 ± 1.07, *p* = 0.002), and 6-month (−2.50 ± 1.15, *p* = 0.029) follow-ups. The WOMAC function score improved continually from baseline at 1-month and 3-month follow-ups but not at 6-month follow-up (Table 3 and Figure 2).

### 3.2. HA Group

The mean WOMAC total score was 18.23 ± 1.80 at baseline and 16.16 ± 2.04, 14.93 ± 2.17, and 13.88 ± 1.91 at the 1-, 3-, and 6-month follow-ups, respectively (Table 2). Compared with the baseline, the WOMAC total score did not significantly improve at the 1- (−2.07 ± 1.50, *p* = 0.163) and 3-month follow-ups (−3.30 ± 1.78, *p* = 0.061) but significantly improved at the 6-month follow-up (−4.36 ± 1.56, *p* = 0.005) (Table 3 and Figure 2).

The mean WOMAC pain score was 4.38 ± 0.43 at baseline and 3.21 ± 0.39, 2.88 ± 0.41, and 2.79 ± 0.38, respectively, at the 1-, 3-, and 6-month follow-ups, respectively (Table 2). Compared with the baseline, the WOMAC pain score significantly improved at the 1-month (−1.16 ± 0.37, *p* = 0.002), 3-month (−1.50 ± 0.43, *p* < 0.001), and 6-month (−1.59 ± 0.40, *p* < 0.001) follow-ups (Table 3 and Figure 2).

The mean the WOMAC stiffness score was 1.73 ± 0.21 at baseline and 1.57 ± 0.19, 1.59 ± 0.21, and 1.30 ± 0.16 at the 1-, 3-, and 6-month follow-ups, respectively (Table 2). Compared with the baseline, the WOMAC stiffness score did not significantly improve at the 1-month (−0.16 ± 0.15, *p* = 0.281) and 3-month (−0.14 ± 0.20, *p* = 0.467) follow-ups but significantly improved at the 6-month (−0.43 ± 0.18, *p* = 0.013) follow-up (Table 3 and Figure 2).

The mean WOMAC function score was 12.13 ± 1.27 at baseline and 11.38 ± 1.54, 10.46 ± 1.59, and 9.79 ± 1.44 at the 1-, 3-, and 6-month follow-ups, respectively (Table 2). Compared with the baseline, the WOMAC function score did not significantly improve at the 1-month (−0.75 ± 1.18, *p* = 0.523) and 3-month (−1.66 ± 1.31, *p* = 0.200) follow-ups but significantly improved at the 6-month (−2.34 ± 1.17, *p* = 0.044) follow-up (Table 3 and Figure 2).

### 3.3. PRP vs. HA

When the improvements in WOMAC scores from baseline to 1-month follow-up between groups were compared, no significant differences in WOMAC total (*p* = 0.369), pain (*p* = 0.634), stiffness (*p* = 0.762), and function (*p* = 0.214) scores were found. However, the improvements in WOMAC function and total score in the PRP group were better than those in the HA group (Table 3 and Figure 2).

When the improvements in WOMAC scores from baseline to 3-month follow-up between the groups were compared, no significant differences in WOMAC total (*p* = 0.380), pain (*p* = 0.793), stiffness (*p* = 0.376), and function (*p* = 0.329) scores were observed. However, the improvements in WOMAC stiffness, function, and total score in the PRP group were better than those in the HA group (Table 3 and Figure 2).

When the improvements in WOMAC scores from baseline to 6-month follow-up between groups were compared, no significant differences were observed in WOMAC total (*p* = 0.998), pain (*p* = 0.651), stiffness (*p* = 0.719), or function (*p* = 0.921) scores, and the improvements in all of these scores in the PRP group were similar to those in the HA group (Table 3 and Figure 2).

## 4. Discussion

This is the first prospective, double-blind, randomized controlled trial comparing the efficacy of intra-articular single PRP versus novel crosslinked HA (HyajointPlus) for the treatment of early-stage knee OA with robust evidence and a large sample size. The results revealed that the PRP group had significant improvements in WOMAC pain, stiffness, function, and total scores at all follow-up visits, except for the WOMAC stiffness score at 1-month follow-up relative to baseline. The HA group exhibited significant improvement in the WOMAC pain score at all follow-up visits and had a significant improvement in WOMAC stiffness, function, and total scores at the 6-month follow-up relative to baseline. For the PRP group, the improvements in WOMAC pain, function, and total scores improved from baseline to the 1- and 3-month follow-up but did not continually improve at the 6-month follow-up. The WOMAC stiffness continually improved from baseline to 1-, 3-, and 6-month follow-ups. For the HA group, the improvements in WOMAC pain, stiffness, function, and total scores were continually increased from baseline to 1-, 3-, and 6-month follow-ups. However, no significant between-group differences were observed in WOMAC pain, stiffness, function, and total scores at any follow-up visit. Thus, both PRP and HA significantly improved all WOMAC scores at the 6-month follow-up with comparable efficacy.

The severity of knee OA may affect the efficacy of intra-articular HA. A meta-analysis reported that intra-articular HA therapy was associated with significant improvement in pain relief among patients with early–moderate knee OA (K–L grades 1–3) but not patients with end-stage knee OA (K–L grade 4) [31]. This indicates that patients with end-stage knee OA do not respond to HA therapy. Accordingly, we only included patients with early-stage knee OA (K–L grade 1–2).

Patients with knee OA are primarily concerned with pain [32]. Improvement in physical activity has been associated with pain relief [33]. We observed that WOMAC pain scores significantly improved at all follow-up visits in both the PRP and HA groups. The accepted threshold for a minimum clinically important difference (MCID) in the WOMAC score for OA of the lower limbs is a 12% improvement from baseline [34]. In our study, improvements in the WOMAC total score from baseline to the 6-month follow-up were 23.21% and 23.92% in the PRP and HA groups, respectively, both being greater than the MCID. Thus, our results indicated good pain and functional results in both the PRP and the HA groups.

The PRP preparation method may affect the efficacy of intra-articular PRP therapy for knee OA. Several RCTs preparing PRP using a leukocyte-rich technique reported no significant difference in self-reported pain and symptoms between PRP and HA groups [17,35,36]. By contrast, several meta-analyses have reported that PRP prepared using a leukocyte-poor technique was associated with significantly better improvement in self-reported symptoms relative to a HA group, indicating that it is a better choice for treating knee OA (although more evidence is required) [24,37]. PRP injections yield favorable outcomes for knee OA, but the optimal injection number and frequency remain unclear. A meta-analysis reported that multiple injections of PRP for treating knee OA were more effective in improving joint functionality relative to a single injection at 6 months [38]. A study using three injections for treating knee OA reported that the improvement in the WOMAC pain score in the PRP group was significantly better than that in the HA group [39]. However, one injection of PRP is also safe and effective for treating knee OA [40,41,42]. In this study, we adopted single-injection therapy using a leukocyte-poor technique, and we observed that the improvements in WOMAC pain, stiffness, function, and total scores in the PRP group were significant at all follow-up visits. However, the optimal injection regimen remains undetermined and thus a topic for further investigation.

Intra-articular HA injection has long been used to treat knee OA, and most HA preparations require multiple injections. However, differences in HA formulations, origins, concentrations, and dosing regimens mean that the present body of clinical evidence is inconclusive. In our study, we used the novel crosslinked HA (HYAJOINT Plus) as a single-injection therapy. To date, only a few studies have reported the efficacy of HYAJOINT Plus for treating knee OA. Sun et al. indicated that a single injection of HYAJOINT Plus is safe and effective for 6 months in patients with K–L 2 or 3 knee OA [27]. Tuan et al. also concluded that a single injection of HYAJOINT Plus is effective with respect to both subjective and objective indicators for 6 months in patients with K–L 2 or 3 knee OA [28]. A recent study by Sun et al. demonstrated that combined injections of HYAJOINT Plus and PRP are safe and effective for 6 months in patients with K–L 2 knee OA [29]. In this study, we observed that the improvements in WOMAC pain, stiffness, function, and total scores were significant at the 6-month follow-up. The single-injection regimen could reduce the time spent on and the discomfort associated with the injection process, thus offering potential benefits to patients with knee OA.

Four patients in the HA group developed mild joint swelling, with pain after injection, but it resolved spontaneously within 1–3 days. No infection, allergy, or other severe adverse events were noted during the study.

This study has some limitations. First, the efficacy of PRP versus HA was unclear after 6 months due to our relatively short follow-up period of up to 6 months. A study revealed that PRP and HA treatments yielded significant improvements at a 24-month follow-up [18]. Second, we did not use real-time image guidance during the injection procedure to ensure that the location of the injection site was accurate; this may affect the efficacy of PRP or HA treatment and possibly induce bias in our comparison between the two groups. Third, pain, stiffness, and functional outcomes were assessed using self-reported questionnaires, and our results lacked objective data, such as those from magnetic resonance imaging. Fourth, we did not include a placebo group. Although a previous meta-analysis revealed a 40–50% reduction in pain with the use of HA compared with a placebo [43], the placebo effect cannot be identified. Fifth, the standard deviation(SD) value of the WOMAC score is relatively large, which shows the disadvantage that the discreteness of the WOMAC score value is not concentrated in our study. More accurate questionnaires or data collection methods may be needed to improve this shortcoming.

## 5. Conclusions

This double-blind RCT provided robust evidence on the efficacy of intra-articular PRP versus HA in the treatment of early-stage knee OA. The results revealed that both PRP and HA can yield significant improvements in WOMAC scores at 6-month follow-up and WOMAC pain scores at 1-, 3-, and 6-month follow-ups without any between-group differences. However, PRP tended to yield greater improvement in WOMAC stiffness, function, and total scores relative to the HA group at 3-month follow-up.

## Figures and Tables

**Figure 1 medicina-58-01028-f001:**
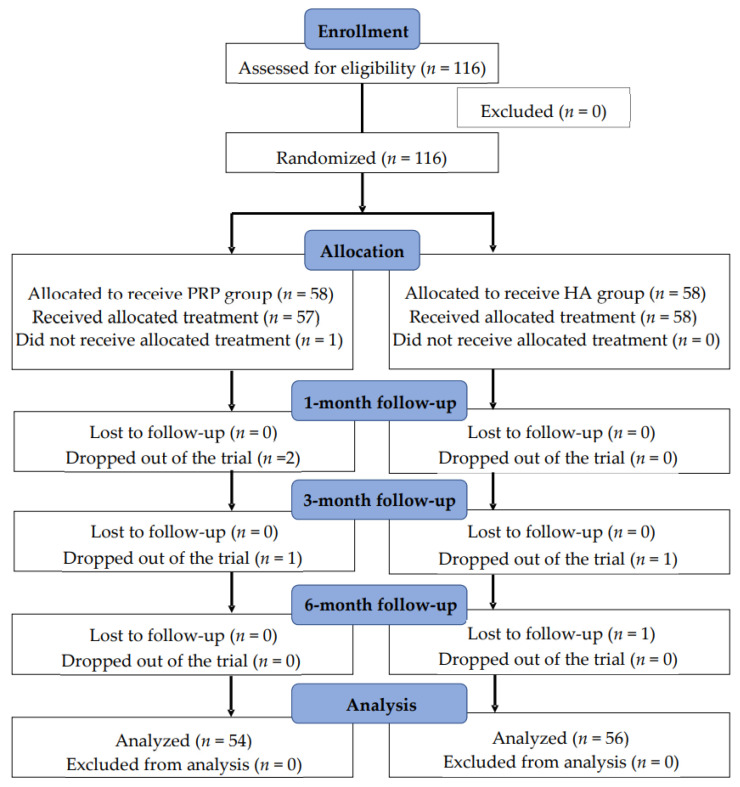
Flow chart of this study.

**Figure 2 medicina-58-01028-f002:**
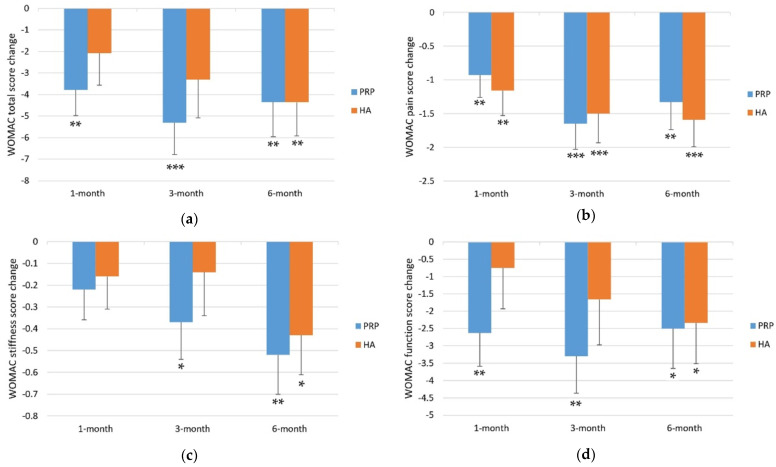
Mean changes in WOMAC scores from baseline to follow-up visits. (**a**) WOMAC total score change; (**b**) WOMAC pain score change; (**c**) WOMAC stiffness score change; (**d**) WOMAC function score change. Error bars indicate one standard error of the mean. Asterisks indicate p-values for the difference from baseline to follow up visits. * *p*-value < 0.05. ** *p*-value < 0.01. *** *p*-value < 0.001.

**Table 1 medicina-58-01028-t001:** Main characteristics of the included patients in the two treatment groups.

	PRP Group (*n* = 54)	HA Group (*n* = 56)	*p*-Value
Age (year)	61.87 ± 5.46	63.00 ± 5.33	0.274 ^a^
Gender			0.445 ^b^
Male, *n* (%)	12 (22.2%)	16 (28.6%)	
Female, *n* (%)	42 (77.8%)	40 (71.4%)	
BMI	24.07 ± 3.35	24.02 ± 2.39	0.937 ^a^
K–L grade			0.055 ^b^
I	29 (53.7%)	40 (71.4%)	
II	25 (46.3%)	16 (28.6%)	
Treatment Side			0.686 ^b^
Right	23	26	
Left	31	30	
WOMAC score			
Pain	4.20 ± 0.42	4.38 ± 0.43	0.776 ^a^
Stiffness	1.89 ± 0.19	1.73 ± 0.21	0.575 ^a^
Function	12.65 ± 1.38	12.13 ± 1.27	0.780 ^a^
Total	18.74 ± 1.85	18.23 ± 1.80	0.844 ^a^

K–L: Kellgren–Lawrence. BMI: body mass index. ^a^ between-group differences were assessed by atwo-sample *t*-test. ^b^ between-group differences were assessed by an *X*^2^ test.

**Table 2 medicina-58-01028-t002:** Mean WOMAC scores at different follow-up visits in the two treatment groups.

	PRP Group (*n* = 54)	HA Group (*n* = 56)	*p*-Value
WOMAC at 1-month follow-up			
Pain	3.28 ± 0.37	3.21 ± 0.39	0.907 ^a^
Stiffness	1.67 ± 0.19	1.57 ± 0.19	0.722 ^a^
Function	10.02 ± 1.15	11.38 ± 1.54	0.483 ^a^
Total	14.96 ± 1.60	16.16 ± 2.04	0.646 ^a^
WOMAC at 3-month follow-up			
Pain	2.56 ± 0.32	2.88 ± 0.41	0.545 ^a^
Stiffness	1.52 ± 0.16	1.59 ± 0.21	0.791 ^a^
Function	9.35 ± 1.03	10.46 ± 1.59	0.562 ^a^
Total	13.43 ± 1.42	14.93 ± 2.17	0.567 ^a^
WOMAC at 6-month follow-up			
Pain	2.87 ± 0.35	2.79 ± 0.38	0.872 ^a^
Stiffness	1.37 ± 0.17	1.30 ± 0.16	0.774 ^a^
Function	10.15 ± 1.12	9.79 ± 1.44	0.843 ^a^
Total	14.39 ± 1.56	13.88 ± 1.91	0.836 ^a^

WOMAC: the Western Ontario and McMaster Universities Index. PRP: platelet-rich plasma. HA: hyaluronic acid. ^a^ the difference between the groups was assessed by a two-sample *t*-test.

**Table 3 medicina-58-01028-t003:** Changes in WOMAC scores from baseline to follow-up visits in the two treatment groups.

	PRP Group (*n* = 54)	*p*-Value ^a^ (Pre vs. Post)	HA Group (*n* = 56)	*p*-Value ^b^ (Pre vs. Post)	*p*-Value ^c^
WOMAC at 1-month follow-up					
Pain	−0.93 ± 0.33	0.005 **	−1.16 ± 0.37	0.002 **	0.634
Stiffness	−0.22 ± 0.14	0.107	−0.16 ± 0.15	0.281	0.762
Function	−2.63 ± 0.96	0.006 **	−0.75 ± 1.18	0.523	0.214
Total	−3.78 ± 1.20	0.001 **	−2.07 ± 1.50	0.163	0.369
WOMAC at 3-month follow-up					
Pain	−1.65 ± 0.38	<0.001 ***	−1.50 ± 0.43	<0.001 ***	0.793
Stiffness	−0.37 ± 0.17	0.026 *	−0.14 ± 0.20	0.467	0.376
Function	−3.30 ± 1.07	0.002 **	−1.66 ± 1.31	0.200	0.329
Total	−5.31 ± 1.48	<0.001 ***	−3.30 ± 1.78	0.061	0.380
WOMAC at 6-month follow-up					
Pain	−1.33 ± 0.41	0.001 **	−1.59 ± 0.40	<0.001 ***	0.651
Stiffness	−0.52 ± 0.18	0.004 **	−0.43 ± 0.18	0.013 *	0.719
Function	−2.50 ± 1.15	0.029 *	−2.34 ± 1.17	0.044 *	0.921
Total	−4.35 ± 1.61	0.006 **	−4.36 ± 1.56	0.005 **	0.998

WOMAC: the Western Ontario and McMaster Universities Index. PRP: platelet-rich plasma. HA: hyaluronic acid. ^a^
*p*-values for PRP group. ^b^
*p*-values for HA group. ^c^
*p*-values for the difference between the PRP and HA groups. * *p*-value < 0.05. ** *p*-value < 0.01. *** *p*-value < 0.001.

## Data Availability

The data presented in this study are available on request from the corresponding author. The data are not publicly available because of confidentiality issues.

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
