# Peer review of "Comparing the Efficacy of Intra-Articular Single Platelet-Rich Plasma(PRP) versus Novel Crosslinked Hyaluronic Acid for Early-Stage Knee Osteoarthritis: A Prospective, Double-Blind, Randomized Controlled Trial"

_medicina, 2022, doi:10.3390/medicina58081028_

Round 1
Reviewer 1 Report
The following are the points that should be addressed before the paper could be considered for publication:
Major:
· To define the study as “A prospective, double-blind, randomized controlled trial” the CONSORT checklist should be fulfilled with discussion on each criteria
· The Hypothesis should be defined
· The primary outcome should be defined
· Intrarticular injection technique should be described
· SDs are larger than mean values – should be explained and discussed; instead of SDs, SEMs should be given for better understanding
Minor:
· Line 49 : “under cartilage” change to subchondral
· Line 55: “last choice” – rephrase
· Lines 58, 65 : PRP and HA – provide full terminology before the abbreviation
· Lines 67 – 68 : “In individuals with knee OA, HA has a lower molecular weight and concentration; thus, intra-articular HA injection is widely adopted to treat knee OA” - provide the reasoning
· Lines 122, 125, 144 - Aeon Acti-PRP, HYAJOINT Plus, G*Power software (v.3.0.10) (3 mL, SciVision Biotech – provide the manufacturers details (addresses and names)
Reviewer 2 Report
Maybe a longer follow up is necessary for see some differences between treatments.
Why only one injection of L-PRP if is demonstrated that best outcomes are after a three infiltration protocol?
Round 2
Reviewer 1 Report
The following points were not addressed sufficiently :
Point 1 - To define the study as “A prospective, double-blind, randomized controlled ” the CONSORT checklist should be fulfilled with a discussion on each criterion in the text
point 3 – the technique of injections means that the description of syringe, needle, and exact point of injection should be given
POINT 4 - this study intends to indicate a conclusion on how well the study results truly represent the expected results in the entire population. For this purpose, the SEMs and not SDs should be presented
Point 6 – "final solution" is also not sufficient and not correct terminology
Point 8 – "intra-articular HA is reduced both in concentration and molecular weight in literatures" – the reasoning for injection of high or low MW HA should be discussed extensively. "In literatures" – should be rephrased
Round 3
Reviewer 1 Report
All the review suggestions are addressed. The paper is ready for publication
This manuscript is a resubmission of an earlier submission. The following is a list of the peer review reports and author responses from that submission.